# Bioactivity-Guided Isolation and Antihypertensive Activity of *Citrullus colocynthis* Polyphenols in Rats with Genetic Model of Hypertension

**DOI:** 10.3390/medicina59101880

**Published:** 2023-10-23

**Authors:** Neelam Iftikhar, Abdullah Ijaz Hussain, Tabinda Fatima, Bader Alsuwayt, Abdullah K. Althaiban

**Affiliations:** 1Natural Product and Synthetic Chemistry Laboratory, Department of Chemistry, Government College University Faisalabad, Faisalabad 38000, Pakistan; neelamiftikhar026@gmail.com; 2Central Hi-Tech Laboratory, Government College University Faisalabad, Faisalabad 38000, Pakistan; 3Department of Pharmaceutical Chemistry, College of Pharmacy, University of Hafr Al Batin, Hafar Al Batin 39524, Saudi Arabia; tabinda@uhb.edu.sa; 4Department of Pharmacy Practice, College of Pharmacy, University of Hafr Al Batin, Hafar Al Batin 39524, Saudi Arabia; balsuwayt@uhb.edu.sa (B.A.); akalthaiban@uhb.edu.sa (A.K.A.)

**Keywords:** quercetin, ferulic acid, pulse wave velocity, polyphenols, gradient elusion, pulse pressure, SHR

## Abstract

*Background and Objectives*: *Citrullus colocynthis* belongs to the Cucurbitaceae family and is a wild medicinal plant used in folk literature to treat various diseases. The purpose of the current study was to explore the antihypertensive and antioxidant potentials of *Citrullus colocynthis* (CC) polyphenol-rich fractions using a spontaneous hypertensive rat (SHR) model. *Materials and Methods*: The concentrated aqueous ethanol extract of CC fruit was successively fractioned using solvents of increasing polarity, i.e., hexane, chloroform, ethyl acetate and n-butanol. The obtained extracts were analyzed for total phenolic content (TPC), total flavonoid content (TFC) and total flavonol content (TOF). Moreover, the CC extracts were further evaluated for radical scavenging capacity using 2,2-diphenyl-1-picrylhydrazyl (DPPH) and 2,2-azino-bis-3-ethylbenzothiazoline-6-sulfonic acid (ABTS) assays and antioxidant activity using inhibition of linoleic acid peroxidation and determination of reducing potential protocols. The phytochemical components were characterized by HPLC–MWD–ESI–MS in positive ionization mode. *Results*: The results showed that ethyl acetate fraction (EAF) exhibited a higher content of phenolic compounds in term of TPC (289 mg/g), TFC (7.6 mg/g) and TOF (35.7 mg/g). EAF showed higher antioxidant and DPPH and ABTS scavenging activities with SC_50_ values of 6.2 and 79.5 µg/mL, respectively. LCMS analysis revealed that twenty polyphenol compounds were identified in the EAF, including phenolic acids and flavonoids, mainly myricetin and quercetin derivatives. The in vivo antihypertensive activity of EAF of CC on SHR revealed that it significantly decreased the mean arterial pressure (MAP), systolic blood pressure (SBP), diastolic blood pressures (DBP) and pulse pressure (PP) as compared to normal and hypertensive control groups. Moreover, EAF of CC significantly reduced the oxidative stress in the animals in a dose-dependent manner by normalizing the levels of superoxide dismutase (SOD), malondialdehyde (MDA), reduced glutathione (GSH), nitric oxide (NOx) and total antioxidant capacity (TAC). Furthermore, the treatment groups, especially the 500 mg of EAF per kg body weight (EA-500) group, significantly (*p* ≤ 0.05) improved the electrocardiogram (ECG) pattern and pulse wave velocity (PWV). *Conclusion:* It was concluded that the EAF of CC is a rich source of polyphenols and showed the best antioxidant activity and antihypertensive potential in SHR.

## 1. Introduction

Hypertension (HTN) is defined as a medical disorder that results in a persistent increase in blood pressure (>130/80 mm Hg), exerted against the walls of arteries [1,2]. Hypertension is a chronic disease known as an infamous silent killer without prominent symptoms, signs or warnings [3]. According to the World Hypertension League, over 50% of the hypertensive population is unaware of their conditions [4,5,6]. Hypertension, along with other cardiovascular risk factors, has been established as the cause of approximately 70–76% of strokes and 83–89% of ischemic heart disease in the world [4,7]. Furthermore, it could be responsible for more critical conditions, including impairing cognitive function, increasing left ventricular mass, arteriosclerosis and atrial fibrillation [6,8]. According to the World Health Organizations (WHO), HTN causes 7.5 million deaths each year, and approximately more than 1 billion people around the world are suffering from hypertension, with an estimated 1.25 billion by 2025 [1,9,10]. The prevalence of HTN is rapidly growing in poor and developing countries, and according to the National Health Survey of Pakistan, 18% of adults between 15 and 45 years and 33% above 45 years are suffering from HTN and stress [11]. 

Risk of hypertension is associated with many factors, including oxidative stress, which account for approximately 90–95% of the total high blood pressure cases [11]. Reactive oxygen species (ROS) and free radicals produced by oxidative stress area cause of vascular damage and endothelial dysfunction, which leads to increased blood pressure [12,13]. To control HTN and high blood pressure (HBP), different non-pharmacological (preventive) and pharmacological (curative) approaches are in practice [1,11]. Pharmacological approaches include the utilization of various antihypertensive medications, including diuretics, beta blockers, calcium channel blockers and angiotensin-converting enzyme inhibitors [1]. However, the adoption of synthetic drugs has presented a number of challenges, including the inaccessibility of drugs and various side effects associated with these drugs [3,4]. Therefore, researchers are focused on finding alternative approaches that should be effective, inexpensive and safe [1]. 

To this end, phyto-bioactive compounds, especially polyphenols, are often considered a good alternative as many have demonstrated hypotensive capacity and have fewer side effects and lower cost [14,15]. Plant-based natural products, especially polyphenols, are natural antioxidants and are very effective against oxidative stress and are therefore used to treat HTN and reduce the incidence of cardiovascular disease [3,7,16,17]. Quertetin, curcumin and resveratrol are important polyphenols that exert vasoprotective effects by increasing nitric oxide production [15]. 

*Citrullus colocynthis* L. Schrad. (*C. colocynthis*) belongs to the Cucurbitaceae family and grows wild in hot arid regions, mainly in India, Pakistan, Sri Lanka, Bangladesh and desert areas of Saudi Arabia [18,19]. *C. colocynthis* is famous in folk literature for the treatment of digestive tract diseases, diabetes and arterial HTN [20,21]. The plant also has a history for its folk use against cancer and purgative action on the gastro-intestinal tract [22]. In recent literature, some reports on the phytochemistry and various biological activities, such as antioxidant, anticancer, anti-inflammatory, cardioprotective and anti-diabetic activities, of *C. colocynthis* extracts have been published [23,24,25,26]. *C. colocynthis* contains various bioactive compounds, including flavonoids (kaempferol, quercetin, myricetin, etc.) and polyphenols (chlorogenic acid, caffeic acid, etc.), that are potential antioxidants and cardioprotectives [12,15,27]. The antihypertensive activity of polyphenol-rich extracts of *C. colocynthis* fruits against the diseased model is not studies frequently.

Therefore, in the present study, we aimed to determine the effect of polyphenol-rich fractions of *C. colocynthis* fruits, prepared using various solvents, for antioxidant and antihypertensive activities.

## 2. Materials and Methods

### 2.1. Reagents, Reference Compounds and Chemicals

Solvents for the mobile phase of chromatography were LCMS-grade and procured from Merck (Darmstadt, Germany), and water was procured from a Milli-Q system (Millipore, Bedford, MA, USA). All the phenolic acid and flavonoid standards and reagents used in this study were procured from Sigma Chemical Co. (St. Louis, MO, USA). Other chemicals used in the in vitro analysis were procured from Sigma-Aldrich (St. Louis, MO, USA). For ex vivo analysis, the kits were procured from Nanjing Jiancheng, Bioengineering Institute, Nanjing, Jiangsu, China.

### 2.2. Plant Materials

Fresh, wildly grown *C. colocynthis* fruits were collected in the summer from the Cholistan Desert, South Punjab, Pakistan. The species of the CC was identified by the Taxonomist, Department of Botany, University of Agriculture Faisalabad, Pakistan and authenticated with a voucher (voucher specimen code, *C. colocynthis* (2734), University of Agriculture, Faisalabad, Pakistan). The plant material was dried in a hot-air dehydrator and ground to a fine powder (80-mesh) using a grinder and submitted for extraction. 

### 2.3. Sample Preparation

Multiple batches of ground plant material (300 g) were macerated and extracted with 700 mL of absolute ethanol in a Soxhlet unit (1-L capacity) for 16–18 h, as reported previously [27]. The extracts were then combined, filtered and concentrated under reduced pressure using a rotary evaporator (Rotavapor R-300, BṺCHI, Labortechnik AG, Flawil, Switzerland). The dried, crude concentrated extract was weighed to calculate the yield of crude ethanol extract. The obtained extract was dissolved in aqueous ethanol (1:10 *w*/*v*) and subjected to a liquid–liquid extraction in a separating funnel using solvents of increasing polarity, i.e., hexane, chloroform, ethyl acetate and n-butanol, to separate the hexane fraction (HEF), chloroform fraction (CHF), ethyl acetate fraction (EAF) and butanol fraction (BUF), respectively (Figure 1). The remaining phase was considered as the 5th fraction, i.e., aqueous ethanol fraction (AEF). The solvents were again removed under reduced pressure using the rotary evaporator, and yields were calculated.

### 2.4. Evaluation of In Vitro Antioxidant Activity

#### 2.4.1. Determination of Total Phenolic, Flavonoid and Flavonol Content

The total phenolic content (TPC) of CC fractions was determined using the Folin–Ciocalteu method as reported earlier, with the modification that 20 mg of extracts was used [28]. The absorbance of gallic acid solutions (0.025–0.400 mg/mL) and extracts were measured at 755 nm using a UV–Vis double-beam spectrophotometer (Lambda 25, Perkin-Elmer, Shelton, CT, USA). The TPCs were calculated using a gallic acid calibration curve (y = 6.2507χ + 0.0344; R^2^ = 0.9992), and the results are reported in mg gallic acid equivalent (GAE) per gram (mg/g) of dry material. The total flavonoid content (TFC) of CC fractions was determined following the reported procedure [28]. The absorbance of catechin solutions (0.025–0.150 mg/mL) and extracts were recorded at 510 nm and TFC was calculated using a catechine calibration curve (y = 2.3727χ + 0.0048; R^2^ = 0.9986), and results are reported in mg catechin equivalent (CE) per gram (mg/g) of dry plant material. The flavonol (FOL) content of CC fractions was determined following the reported method [28]. Briefly, 10 mg of extract was mixed with 1 mL of aluminum trichloride (2%, w) and 1.5 mL of 5% sodium acetate solution (w). The mixture was then incubated at 25 °C for 3 h. The absorbance of rutin solution (0.025–0.15 mg/mL) and extracts were measured at 440 nm. The FOL content was calculated using a rutin calibration curve (y = 8.1339χ − 0.0014; R^2^ = 0.9991), and results are reported in mg rutin equivalent (RE) per gram (mg/g) of dry plant material.

#### 2.4.2. DPPH Radical Scavenging Assay

Various concentrations of CC fractions (1–100 μg/mL) were prepared for the evaluation of DPPH-free radical scavenging activity using the reported method [29]. An amount of 1 mL of each fraction solution was mixed with 1 mL of 90 µM methanol solution of DPPH and absorbance was recorded at 517 nm after 30 min incubation at room temperature. The DPPH solution was taken as a blank and BHT and BHA were used as positive controls for comparison. The radical scavenging in percentage was calculated using the given formula.
DPPH Radical Scavenging (%) = (Absorbance of DPPH solution − Absorbance of sample solution)/(Absorbance of DPPH solution) × 100

The extract concentration providing 50% scavenging of DPPH radical (SC_50_) was calculated from the graph-plotted scavenging percentage against extract concentration.

#### 2.4.3. ABTS Scavenging Assay

The radical scavenging activity was also determined using the stable cation radical ABTS, as reported [30]. The ABTS radical was synthesized by mixing the ABTS solution (7 mM) with 13.2 mg of potassium persulfate for 16 h. The solution was refrigerated and then diluted to reach the absorbance of 0.70 at 734 nm. An amount of 100 μL of CC fractions (10–10,000 μg/mL) was mixed with 1.9 mL of ABTS solution and incubated at room temperature for 10 min. The absorbance was recorded at 734 nm. The same concentrations of BHT and BHA were used as a positive control for comparison. The ABTS scavenging was calculated by the following formula.
ABTS Inhibition (%) = (Absorbance of ABTS solution − Absorbance of sample solution)/(Absorbance of ABTS solution) × 100

Extract concentration providing 50% scavenging of the ABTS initial concentration (SC_50_) was calculated by plotting a graph between scavenging percentage against con-centration of extract.

#### 2.4.4. Inhibition of Linoleic Acid Peroxidation

The antioxidant activity of CC fractions was also determined by measuring the inhibition of linoleic acid peroxidation using the extract concentration, as reported [31], with modification. One negative (emulsion only) and two positive (BHT and BHA) controls were also used to calculate the results. The absorbances of various samples were recorded at 500 nm at the start and after 180 h. The inhibition of linoleic acid peroxidation was calculated using the given formula.
Inhibition (%) = 100 − [((Absorbance increase in sample)/(Absorbance increase in negative control)) × 100]

#### 2.4.5. Determination of Reducing Power

The antioxidant of CC fractions, in term of reducing power, was determined according to the procedure reported earlier [32]. Briefly, 0.625–10.0 mg of the sample was mixed as a reaction mixture (2500 µL of 1.0% potassium ferricyanide and 2500 µL of 0.2 M sodium phosphate buffer of pH 6.6). The mixture was incubated at 50 °C for 20 min and centrifuged at 980× *g* for 10 min at 5 °C in a refrigerated centrifuge (Megafuge 8R; Thermo Scientific, Thermo Electron, Osterode, Germany) after adding 2500 µL of 10% trichloroacetic acid. An amount of 5.0 mL of the upper layer was diluted with 5.0 mL of distilled water and mixed with 1000 µL of 0.1% ferric chloride. The absorbance was recorded at 700 nm and plotted against concentration.

### 2.5. Liquid Chromatography–Mass Spectrometry (LC–MS) Analysis

The Agilent 1260 infinity II LC System (Agilent, Santa Clara, CA, USA), equipped with a gradient-model binary pump (G7112B), 1260 auto sampler (G7129A) and 1260 multi-wavelength UV/Vis detector (G7165A) coupled with MS (ESI source), column oven and degasser systems, was used for the analysis. The ethyl acetate fraction of CC was analyzed using LC–MS in positive mode as described [33], with slight modification. Ten microliter of the filtrate was injected and separated in a poroshell 120 C_18_ column (150 × 4.6 mm internal diameter, 2.7 µm particle size). The run was performed at a flow rate of 1.0 mL/min. The temperature of the column was fixed at 30 °C. Non-linear gradient (acetonitrile:methanol (70:30) from 15–80% and water with 0.1% formic acid were used. The MWD detector was set at 250, 270, 290, 310, 330, 350, 370 nm 1.2 nm resolution and 10 points/sec sampling rate. The ESI source in positive mode was selected, with a probe temperature of 600 °C, 10 mL/min flow rate and 45 psi nebulizer gas. The MS was set with a total scan range from 100 to 1250 *m*/*z* and a fragmentation voltage of 130 V. Data processing was performed using bulletin control panel software (Agilent Open Lab Data Analysis-Build 2.204.0.661) and retention time (tR), elution order, fragmentation pattern and Base *m*/*z* were noted to identify the compounds. Meanwhile, quantification was based on the standard addition method and the calibration curve was constructed by plotting concentration against the increase in the peak area.

### 2.6. In Vivo Activities

#### 2.6.1. Animals

Adult, male Wistar Kyoto (WKY) rats and Spontaneous Hypertensive rats (SHR) (weighing approximately 140–160 g) were collected from the Animal Breeding House, Department of Pharmaceutical Sciences, Government College University Faisalabad, Pakistan. Animals were housed in a standard polypropylene cage (16.0 × 13.5 × 6.25 inch) under constant temperature (25 ± 3 °C) and humidity (65 ± 6%). The healthy and active rats were distributed in a random manner, and 6 rats were kept in one cage. All in vivo studies were performed after the approval of the Institutional Review Board for Animal Studies (Study No 19680/IRB No 680), Government College University Faisalabad, Pakistan, and were carried out in accordance with the provided guidelines.

#### 2.6.2. Acute Toxicity Study

The study was performed according to the guidelines of the Organization for Economic Co-operation and Development (ORECD), 423-Protocol. No toxic symptoms or mortality was observed until the second week of the period, with all potent CC fractions up to 2250 mg/kg body weight dose.

#### 2.6.3. Experimental Design for Antihypertensive Study

The distributed SHR and WKY rats were kept in the animal transit room for one week for acclimatization and were then marked into the following groups (six rats in each group): (1) Normal Control (NC) group (WKY rats); (2) Hypertensive Control (HC) group (SHR); (3) Positive Control (PC) group (SHR received quercetin (10.0 mg/kg body weight/day) for 28 days); (4) EA-250 group; (5) EA-500 group (SH rats received *C. colocynthis* ethyl acetate fraction (250 and 500 mg/kg BW/day for 28 days, respectively)). The rats were provided normal feed (approximately 20 g/rat/day) and water ad libitum. The treatment doses were given to rats orally by gavage. During the study, the rats were continuously monitored for any visible abnormalities. Animals’ body weights and blood pressures were monitored weekly. 

#### 2.6.4. Non-Invasive Blood Pressure Measurement

Weekly systemic blood pressure parameters, including systolic blood pressure (SBP), diastolic blood pressure (DBP) and heart rate (HR), were measured by the tail cuff method using CODA^®^ (Kent Scientific Corporation, Torrington, CT, USA) containing volume pressure recording (VPR) and occlusion cuff (o-cuff) [34,35]. Rats were acclimatized and trained to stay in a restrainer for 30 min before the actual pressure measurements. Ten consecutive readings were recorded in each session per rat, and average values were calculated. 

### 2.7. Acute Study 

The vascular study was performed 48 h after the end of the 28-day treatment period. 

#### 2.7.1. Surgical Preparation of Animals and Recording of Data

Surgical procedures were performed according to recognized protocols [35,36,37]. On day 29, animals were fasted overnight (12–14 h), with access to water. On the next day, sodium pentobarbitone, 60 mg/kg BW, was injected into the rats for anesthetization. A tracheotomy was performed and PE250 tubing (Portex, Kent, UK) was placed to facilitate clear airways. The left jugular vein was cannulated (PE50 tubing), and it was used for the administration of supplemental anesthesia doses (sodium pentobarbitone, 15 mg/kg i.v) to maintain the anesthesia level during the experiment. The right carotid artery was catheterized and PE50 tubing was passed to the level of the aortic arch to measure the pulse wave velocity (PVW). The left iliac artery was cannulated and PE50 tubing was passed to the level of the abdominal aorta. All the cannulas were connected with the data-acquisition system (Powerlab, AD Instruments, Colorado Springs, CO, USA) through the recording system via Quad Amp using Chart Pro (V.5.5) software. The urinary bladder was pulled up, a small cut was made and a PE10 cannula (Portex, UK) was inserted to allow free passage of urine from the kidney during the experiment. After completion of the surgical procedures, the animals were allowed to stabilize for 1 h before recording the data.

#### 2.7.2. Mean Arterial Pressure Measurement

Direct mean arterial pressure (MAP) was determined from measurements made by the aortic arch pressure transducer and was calculated using built-in software (Chart v5) (Figure 2) as reported [38]. 

#### 2.7.3. Measurement of Pulse Wave Velocity

Pulse wave velocity (PWV) was measure as reported [39,40]. PWV is calculated by dividing “d” (distance travel by a wave) by “t” (time taken by a wave), as presented in the formula and expressed in meter per second as reported (Figure 2). Multiple measurements were recorded, and the data were reported as an average: PWV = (Propagation distance (d))/(Propagation time (t))

#### 2.7.4. Measurement of Electrocardiogram (ECG)

Under the anesthesia, ECG was recorded with lead II position by digital physiograph (BIOPAC, Santa Barbara, CA, USA) [39,41].

### 2.8. Biochemical Investigations

#### 2.8.1. Determination of Oxidative Stress Parameters and Antioxidant Enzyme Activities

At the end of the experiment, blood was collected from the right carotid artery and centrifuged at 3500 rpm for 10 min. The clear layer of plasma was collected and stored at −70 °C until biochemical investigation. The oxidative status of the rats from all the groups was determined using the following assays: measurement of malondialdehyde (MDA), reduced glutathione (GSH), superoxide dismutase (SOD), nitric oxide (NOx) and total antioxidant capacity (TAC) levels, as reported [42,43]. All the reagents were mixed with the plasma as per the instructions of the kit manual, and the absorbance was recorded using a spectrophotometer. 

#### 2.8.2. Collection of Heart

The hearts of all the rats were taken after careful cleaning of blood clots and connective tissues before washing with normal saline. The weight of each heart was recorded. The heart index (HI) was calculated using the following formula.
HI (%) = (Heart weight (g))/(Rat Weight (g)) × 100

### 2.9. Statistical Analysis

All the in vitro analyses were repeated three times, and the data are reported as mean value ± standard deviation. For in vivo analysis, 6 rats were taken from each group, and the results are reported as mean ± standard deviation. To compare the differences between values, one-way Analysis of Variance (ANOVA) followed by Bonferroni/Dunnett (all mean) post hoc test were applied using statistical software (Statistica10.0; Stat Sift Inc., Tulsa, OK, USA) and SPSS-16.0 (IBM, SPSS Inc., Chicago, IL, USA). Probability values of *p* ≤ 0.05 were considered significantly different.

## 3. Results

### 3.1. Extract and Fractions Yields

The ethanol extraction of *C. colocynthis* fruit provided a sticky extract that showed 18.9 g/100 g of the dry fruit material. The liquid–liquid fractionation of the crude extract with various solvents of different polarities is presented in Table 1. The AEF represented the maximum yield (58.6 g/100 g) followed by BUF (16.5 g/100 g), HEF (3.7 g/100 g), EAF (2.9 g/100 g) and CHF (2.3 g/100 g). The significant (*p* ≤ 0.05) differences in the yield of various CC fruit fractions might be due to the variation in the affinity of compounds with the solvents. 

### 3.2. In Vitro Antioxidant Activity

#### 3.2.1. TPC, TFC and TOF Content

Concentrations of TPC, TFC and FOL from CC fractions are shown in Table 1. The EAF contained the highest TPC (289.4 mg/g), followed by CHF (172.3 mg/g), BUF (48.7 mg/g), AEF (26.8 mg/g) and HEF (12.0 mg/g), measured as gallic acid equivalent (GAE). The amounts of TFC from the EAF, CHF, BUF, AEF and HEF were found to be 7.6, 3.5, 2.5, 2.1 and 0.6 mg/g of dry plant material, measured as catechin equivalent (CE), respectively. 

Similarly, the amount of FOL content of EAF was 35.7 mg of rutin equivalent (RE) per gram (mg/g) of dry fruit material, followed by CHF, BUF, AEF and HEF. ANOVA showed significant (*p* ≤ 0.05) variation in the TPC, TFC and TOF content amount of different fractions. 

#### 3.2.2. Radical Scavenging Capacity and Antioxidant Activity

The DPPH and ABTS radical scavenging capacities of the CC fractions were determined, and the results in term of concentration that scavenge/inhibit 50 percent radicals (RC_50_/IC_50_) are presented in Table 1. EAF showed the highest DPPH radical scavenging activity among all the CC fractions, i.e., SC_50_, 6.2 μg/mL, while the hexane fraction showed the least activity, i.e., SC_50_, 115.9 μg/mL. The chloroform fraction (CHF) showed SC_50_, 17.3 μg/mL, followed by BUF (SC_50_, 22.4 μg/mL) and AEF (SC_50_, 53.7 μg/mL). Similarly, all the CC fractions exhibited appreciable ABTS radical scavenging activity and the IC_50_ values ranged from 79.5 to 9782.6 μg/mL. The EAF showed the best activity (IC_50_, 76.5 μg/mL), whereas the HEF exhibited the lowest, which might be due to their low phenolic constituents as compared to other fractions. When compared with the synthetic antioxidants, BHT and BHA showed the best radical scavenging activities against DPPH and ABTS radicals. The ANOVA results showed significant (*p* ≤ 0.05) variation in the DPPH and ABTS radical scavenging potentials of all the CC fractions.

Inhibition of linoleic acid peroxidation was also used to assess the antioxidant activity in term of percent inhibition of lipid peroxidation by different CC fractions, and the results are summarized in Table 1. Different fractions showed different inhibition of linoleic acid peroxidation, and the results are 28.1 to 83.1%. Among the various samples, EAF and CHF were the most potent fractions, inhibiting linoleic acid peroxidation by 83.1% and 79.9%, respectively, which is comparable with BHT and BHA, having inhibition of peroxidation of 88.4 and 89.8%, respectively. One-way ANOVA exhibited significant (*p* ≤ 0.05) variation in the percent inhibition of peroxidation of linoleic acid among all the CC fractions. 

Figure 3 shows the reducing potential of different CC fractions. The results showed an increase in reducing activity with an increase in extract concentration, and the best reducing potential (higher absorbance) was observed from EAF, followed by CHF and HEF. Synthetic antioxidant BHA and BHT showed the highest absorbance, implying the highest reducing potential. 

### 3.3. LC–MS Analysis of Polyphenols

The chemical composition of the ethyl acetate fraction was analyzed by HPLC–MWD–MS, and the results revealed the identification of 20 compounds from the ethyl acetate fraction of CC (Figure 4). Different amounts (mg/100 g of dry plant material) of nine phenolic acids, including gallic acid, p-hydroxy benzoic acid, caffeic acid, p-coumeroylquinic acid, chlorogenic acid, vanillic acid, syringic acid, sinapic acid and ferulic acid, and nine flavonoids, including epicatechin, hesperidin, resveratrol, rutin, isoquercetin, myricetin derivatives, quertetin derivative and eriodictyol-7-O-rutinoside were detected (Table 2). Two glycosides, kaempferol-3-glucoside and apigenin glucoside, were also detected (Table 2). Rutin, chlorogenic acid, sinapic acid, syringic acid, myricetin-3-O-glucuronide, vanillic acid and kaempferol-3-glucoside were found to be major compound (>5.00 mg/100 g of dry plant material). Rutin was also found to be a major compound (27.98 mg/100 g of dry plant material), followed by chlorogenic acid (18.93 mg/100 g of dry plant material), sinapic acid (14.51 mg/100 g of dry plant material) and syringic acid (10.12 mg/100 g of dry plant material).

### 3.4. In Vivo Antihypertensive and Antioxidant Activities

#### 3.4.1. Acute Oral Toxicity

With a dose of 2500 mg/kg body weight, no deaths were observed among the rats throughout the 14-day observation period. During this observation period, the animals did not show any variations in general appearance or activity.

#### 3.4.2. Body Weight and Heart Weight Measurements 

The initial and final average body weights of the control and treatment groups are summarized in Table 3. There were no significant differences (*p* > 0.05) in the initial and final body weight (BW) of different groups, although the final weights of all the groups increased. Heart weight and the ratio of heart weight and body weight (HW/BW) were observed as significantly (*p* ≤ 0.05) increased in the HC group. Treatment groups EA-250 and EA-500 recovered the abnormal HW/BW rations. 

#### 3.4.3. Blood Pressure and Pulse Wave Velocity Recordings

Systolic blood pressure, DBP, MAP, PP, HR and PWV are presented in Table 4. As indicated, the mean arterial pressure in SHR (154.2 mmHg) was significantly higher (*p* ≤ 0.05) than in WKY rats (108.9 mmHg), showing the hypertension level in SHR. The elevated MAP is decreased with the treatments in all the treatment groups. 

Doses of 250 and 500 mg EAF of *C. colocynthis* to SHR decreased the mean arterial pressure in a dose-dependent manner, i.e., 128.3 and 119.8 mmHg, respectively, and the effect was found to be significant (*p* ≤ 0.05). The MAP of SHR was also decreased in the quercetin-treated (PC) group, which is 121.2 mmHg.

The systolic blood pressure of SHR in the HC group (179.4 mmHg) was significantly higher (*p* ≤ 0.05) than in the WKY rats in the NC group (128.1 mmHg). This difference in systolic blood pressure was significantly decreased by the treatment of quercetin in the PC group (140.3 mmHg). Treatment of SHR in the EA-250 and EA-500 groups significantly decreased (*p* ≤ 0.05) the systolic blood pressure. The DBP of the HC group (138.6 mmHg) was significantly higher (*p* ≤ 0.05) when compared with the NC group (92.0 mmHg) (Table 4). However, it was significantly decreased (*p* ≤ 0.05) in all the treatment groups, but not up to the levels of DBP of WKY rats in the NC group. 

The changes in the PP of different groups of the SHR model are shown in Table 4. The pulse pressure of the HC group (41.0 mmHg) was significantly higher (*p* ≤ 0.05) than WKY rats in the NC group (33.9 mmHg). Treatment with quercetin in the PC group and ethyl acetate fraction of *C. colocynthis* in the EA-250 and EA-500 groups significantly recovered the PP, i.e., 36.1, 38.2 and 35.8 mmHg, respectively. The pulse pressure of the EA-500 group was not significantly (*p* > 0.05) different compared with the PP of WKY rats in the NC group, showing the effectiveness of the ethyl acetate fraction of CC. Table 4 shows the pulse wave velocity (PWV) of the NC, HC and treatment groups. PWV was significantly (*p* ≤ 0.05) higher in the HC group (7.12 m/s) compared with the NC group (5.33 m/s). Treatments with quercetin and 500 mg/kg ethyl acetate fraction of CC significantly recovered the increased PWV. The PWV of the PC and EA-500 groups was found to be 5.89 and 5.80 m/s, respectively. As shown in Table 4, no significant (*p* > 0.05) changes were observed in the HR of the treatment groups with the NC and HC groups. 

#### 3.4.4. Electrocardiogram (ECG) Patterns

The ECG patterns of all rat groups under study are provided in Figure 5. The weakening of R-waves and abnormal ST segment elevation were observed in the SHR of the HC group. The treatment groups, especially the PC and EA-500 groups, demonstrated a significantly (*p* ≤ 0.05) improved ECG pattern, as shown in Figure 5.

#### 3.4.5. Oxidative Stress Parameters 

Table 5 shows the effect of EAF of CC on oxidative parameters. As shown in the table, a significant reduction (*p* ≤ 0.05) in the values of SOD, GSH, NOx and TAC and an increment in the values of MDA were detected in the HC group. This showed the presence of stress in SHR in the HC group. All the treatment groups effectively counter the oxidative stress and normalize the oxidative stress parameters. 

In the MDA assay, there was an increase in the concentration of MDA in the HC (5.79 nmol/L) group of 52.93% from that of the normal control (2.81 nmol/L) group. Treatment with EA (250 and 500 mg/kg) produced a significant (*p* ≤ 0.05) decrease in the elevated levels of MDA by 32.99% and 51.25%, respectively, compared with the HC group. Values of GSH, SOD and NOx declined in the disease control group (HC) compared with the values in the NC group by 23.8%, 16.2% and 40.1%, respectively. These results prove that the animals of the HC group are affected by oxidative stress (Table 5). The values of GSH were 163.2, 124.3, 159.7, 142.9 and 161.0 mg/L in the rats of the NC, HC, PC, EA-250 and EA-500 groups, respectively. The treatment groups, especially the PC and EA-500 groups, recovered the decreased GSH levels that are close to the WHY rats in the NC group. GSH value is quite high in the EAF-500 group, which represents a more protective effect (161.01 mg/L) by the EAF against oxidative stress. Similarly, SOD level in rats of the EA-500 group was 135.7 NU/mL, which is better than in the PC group (133.9 NU/mL), showing the recovery of decreased SOD level of SH rats in the HC group. Likewise, treatment groups reverse the depleted NOx levels of SHR in the PC, EA-500 and EA-250 groups, and the recovered levels were close to the WKY rats in the NC group. Treatment with 500 mg/kg BW ethyl acetate fraction in the EA-500 group showed the best effect and increased the NOx level up to 30.09 mmol/L. In the hypertension control group (HC), a decrease in the value of total antioxidant capacity was observed (1.31 mmol/L) as compared with the normal control group (1.78 mmol/L), i.e., 26.4%. Treatment with ethyl acetate fraction in the EA-250 and EA-500 groups and quercetin in the PC group led to an increase in the total antioxidant capacity by 1.72, 1.79 and 1.73 nmol/L, respectively, which is close to the values of WKY rats in the NC group (1.78 nmol/L). It can be summarized that there was a significant (*p* ≤ 0.05) change in the levels of oxidative parameters in the treatment groups, especially in the EA-500 and PC groups.

## 4. Discussion

The choice of extraction medium or solvent plays a critical role in the extraction of polyphenols from plant materials, and it significantly influences the yield and quality of the extracted polyphenols [44]. The difference in extraction yield of CC in the present study with different solvents can be explained by the variation in the polarities of the solvents to dissolve endogenous compounds [45,46]. Ethanol is generally considered as the preferred solvent for the extraction of polyphenols due to its lower toxicity and effectiveness [44,47]. Some previous studies have also reported the extraction yield of the different parts of *C. colocynthis* fruit. Chekroun et al. (2015) [48] reported that the extraction yield of aqueous extract from *C. colocynthis* fruit was 5.41%, which is greater than the yield of butanol extract of *C. colocynthis* fruit (2.61%). Similarly [49,50] reported that the yield of the methanol extract of *C. colocynthis* fruit was 2.78 g/100 g of dry extract, and also reported that the extraction yield ranged from 0.67% for the crust and 26.4% for the CC pulp. 

Both total phenolic content and total flavonoid content are excellent parameters used in assessing the antioxidant capacity and potential health benefits of natural products [28,47,51]. In TPC assay, Folin–Ciocalteu reagent reacts with polyphenols and forms a complex with absorption maxima near 765 nm [42,51]. Solvent polarity affects the extraction of phenolic compounds [44,45,46]. Polar solvents, including ethanol, ethyl acetate and methanol, tend to extract a broader range of phenolic compounds and can result in higher TPC and TFC values. In particular, ethyl acetate solvent is able to dissolve flavonoid aglycones and phenolic diterpenes [42]. Solvent fractionations of increasing polarities, i.e., hexane to methanol, allow the segregation of phenolic compounds into separate groups [46].

The TPC and TFC results of the present study are in line with the findings of Hsouna and Alayed (2012), who reported the higher polyphenol content from the methanol extract of *C. colocynthis* roots in term of TPC and TFC compared with hexane extract. Several studies [21,22,48] have reported that the butanolic extract of the *C. colocynthis* fruit exhibited higher total polyphenol (221.85 mg gallic acid equivalent/g dry extract) and flavonoid (61.20 mg catechin equivalent/g dry extract) content as compared to the aqueous extract. Similarly, 0.74% of gallic acid equivalents of phenolic compounds and 0.13% of catechin equivalents of total flavonoids were also found from the fresh mass of the CC fruit, as reported in [49]. Ngo et al. (2017) [45] also reported that the mixture of 50% (*v*/*v*) water with ethanol, acetone and methanol showed variation in the TPC. Moreover, Sultana et al. (2009) [46] also reported that the solvent polarity effected the extraction of phenolics from plant material, and higher amounts were extracted in polar solvents. 

The antioxidant activity and free radical scavenging capacity of plant extracts and fractions are correlated with the concentration of polyphenol compounds in the extracts/fractions [51,52]. Polyphenols are an important class of natural compounds that exhibit free radical scavenging activity, inhibition of lipid peroxidation and other antioxidants activities [52]. Therefore, total phenolic and flavonoid content in plants are associated with their antioxidant potential due to the redox properties. Polyphenols are aromatic compounds that have the ability to act as hydrogen atom donors, singlet oxygen quenchers and reducing agents [51,53,54]. 2,2-diphenyl-1-picrylhydrazyl radical (DPPH) is the most widely used radical for in vitro antioxidant evaluation of plant extracts and fractions [51,55]. Percent inhibition of linoleic acid peroxidation assay is a hydrogen-transfer-based assay, frequently used for the evaluation of antioxidant activity of plant extracts [47,56]. Similarly, measurement of reducing potential also represents the antioxidant activity of plant extracts [43]. 

*C. colocynthis* extracts mainly contained phenolic compounds and thus exhibited good antioxidant potential [49,57,58,59]. Contrary to the present findings, Chekroun et al. (2015) [48] reported a lesser DPPH radical scavenging capacity of the butanolic extract of *C. colocynthis* fruit (IC_50_ = 61 µg/mL). However, Hussain et al. (2013) [27] reported stronger DPPH radical scavenging activity by the ethanol extract of *C. colocynthis*, native to Pakistan (IC_50_ = 7.14 µg/mL). In another study, Afzal et al. (2023) [60] reported that the best (28%) DPPH inhibition activity was shown by the methanol extract released by n-butanol (17%) at 0.15 mg/mL extract concentration, whereas chloroform fractions showed 26%. Polyphenols are also known to react with lipid radicals in the termination reaction and transfer hydrogen [61]. The better antioxidant activity of HEF compared with AEF might be due to the fact that the reaction medium in the linoleic acid peroxidation assay is an emulsion of linoleic acid; accordingly, less polar molecules have better availability to lipid radicals than more polar molecules [56]. The methanol extract of *C. colocynthis* is highlighted as having good antioxidant activity in term of lipid peroxidation [62]. Chekroun et al. (2015) [48] also reported that, at a concentration of 3 mg/mL, aqueous and butanolic extracts of *C. colocynthis* fruit showed moderate reducing power, i.e., 0.36 and 0.55, respectively. The results of the present study showed stronger antioxidant and radical scavenging activities than those reported in most studies, which might be due to variations in geographical and ecological conditions and extraction solvents [48]. 

Liquid chromatography–mass spectrometry (LC–MS) is a versatile and powerful analytical technique widely used for the identification and quantification of polyphenols from various plant extracts. Polyphenols are a diverse group of compounds with variable polarities and structures, making LC–MS an ideal choice due to its ability to separate, detect and analyze complex mixtures [63]. Our results are in agreement with the findings of [62], who reported that ethyl acetate extract of *C. colocynthis* roots showed the presence of ferulic acid and caffeic acid as major phenolic acids, followed by p-coumaric acid and vanillic acid. The methanol extract of *C. colocynthis* fruit are well known as a source of gallic acids, caffeic acids, quercetin, benzoic acids, syringic acids, p-coumeric acids and sinapic acids [60]. Flavonoids are important compounds for antioxidant potentials, and ethyl acetate fractions are the richest source of flavonoids [64]. In particular, the strong antioxidant activity of flavonoid glycosides is widely known [65]. LC-based activity profiling in the present study showed the potential application of these compounds. 

The effect of polyphenol-rich extracts on final body weight and heart weight in rats is an important tool due to the potential health implications, particularly in the context of cardiovascular health. An increase in HW/BW ratio in animals represents hypertrophy and inflammation [43]. Treatment with extracts overturned the morphological changes in terms of a decrease in body weight and reduced hypertrophy. Some studies available in the literature report that flavonoid-rich extracts reduced the deposition of collagen and inflammation and thus cardiac hypertrophy [66]. 

Polyphenol-rich extracts have been studied for their potential effect on cardiovascular systems, including heart rate and blood pressure [42,43]. Elevated blood pressure in the hypertensive model of rats may be attributed to increased cardiac output, which might be due to a decrease in the elasticity of blood vessels or blockage or a decrease in the diameter of blood vessels [67]. Pulse wave velocity (PWV) is a measure of arterial stiffness, which is associated with cardiovascular health. A decrease in MAP, SBP, DBP, PP and PWV in the treatments groups can be attributed to the high content of polyphenols present in the ethyl acetate fraction of *C. colocynthis* fruit, and it is well documented that certain polyphenols may have a hypotensive (lowering the blood pressure) effect [43]. Due to the presence of the hydroxyl group on the flavonoid backbone, many flavonoids have a strong antioxidant capacity, which inhibits the production of ROS and leads to increased bioavailability of nitric oxide, a vasodilator [63,68]. Moreover, flavonoids have also been reported to up-regulate the expression of endothelial nitric oxide synthase, which helps to reduce blood pressure [69]. Hence, the decrease in blood pressures, PP and PWV of the *C. colocynthis* fruit treatment group may be attributed to the presence of bioactive phenolics, which might be due to their ACE inhibitory and antioxidant activities [68].

An electrocardiogram (ECG) provides useful information about the electrical activity of the heart. Weakening of the R wave in ECG and elevation of the ST segment are important tools for the diagnosis of myocardial infraction. The ECG may be affected by multiple factors, including coronary ischemia, cardiac overload and oxidative stress [70,71]. Polyphenols with an antioxidant effect may have the potential to mitigate ischemic or repolarization changes in the ST segment, and thus polyphenol-rich fractions of medicinal plants are useful for the pathological recovery in ECG changes [71]. 

Furthermore, oxidative stress is a pathogenic factor in the development of essential hypertension, which can induce inflammation and endothelial dysfunction and increased vascular smooth muscle contraction, and it can eventually lead to hypertension [72,73]. Excessive production of free radicals also causes phospholipid peroxidation, which further enhances damage at cellular levels, and it can be determined by the elevation of MDA content [71]. Overconsumption of GSHr is also due to the damaged myocardium, which protects the tissue proteins from cytotoxic free radicals or from lipid peroxides [71]. Metalloenzyme, SOD, is helpful in the reaction of dismutation of superoxide radicals, and depleted levels of SOD in the serum show the development of oxidative stress [43,74]. In the present study, all treatment groups effectively decreased these pathological changes compared to the HC group, mainly due to their antioxidant potential and high polyphenol content. It has also been pointed out that the mitigation of hypertension by dietary polyphenols is associated with the enhancement of NO production [75]. Flavonoids such as quercetin and metabolites attenuate hypertension through modulation of RAAS, suppressing the expression of NADPH subunits and modulating VSMC contractility [72,75,76]. Previous studies have also reported that treatment with polyphenols and quercetin results in a decrease in blood pressure in hypertensive animals and humans due to a decrease in oxidative stress [77]. Moreover, catechin intake is also associated with lower diastolic and systolic blood pressure [76]. The first report on the antihypertensive effects of quercetin was also from a study carried out on spontaneously hypertensive rats (SHR) [41]. The spontaneously hypertensive rat (SHR) model was selected in the present study because it is the most extensively used animal model to study hypertension [5,78]. 

## 5. Conclusions

The present study explored the antioxidant and antihypertensive potential of *C. colocynthis* fractions. The results showed that ethyl acetate fraction is the richest polyphenol fraction, with higher TPC, TFC and FOL content and maximum in vitro antioxidant and free radical scavenging activities. LCMS analysis of the EAF of CC revealed 20 polyphenol compounds, including rutin, chlorogenic acid, quercetin and myricetin derivatives. In vivo analysis showed that the EAF of CC demonstrated substantial antioxidant and antihypertensive activities in SHR. These results provide good evidence to consider the EAF of CC as potent agents for antioxidant and antihypertensive properties and can be considered as a cardioprotective agents in further studies.

## Figures and Tables

**Figure 1 medicina-59-01880-f001:**
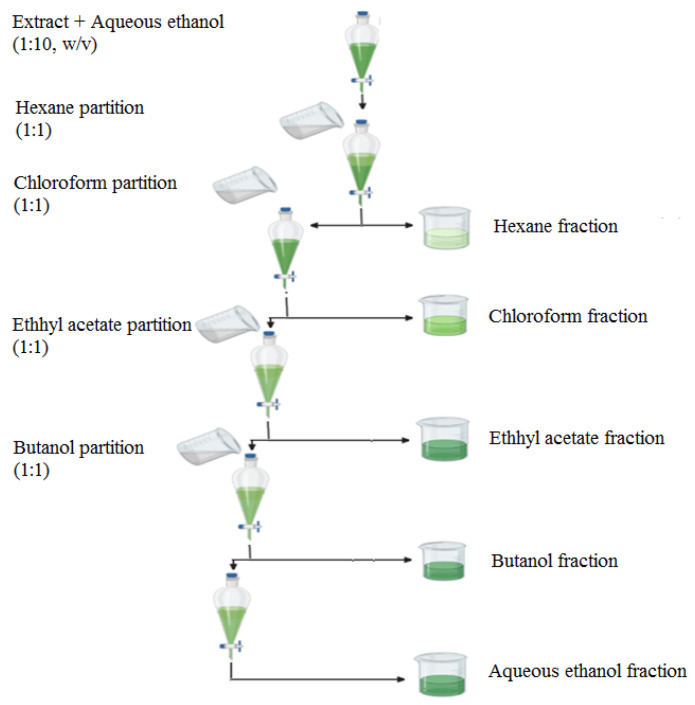
Schematic representation of liquid–liquid extraction in various solvents.

**Figure 2 medicina-59-01880-f002:**
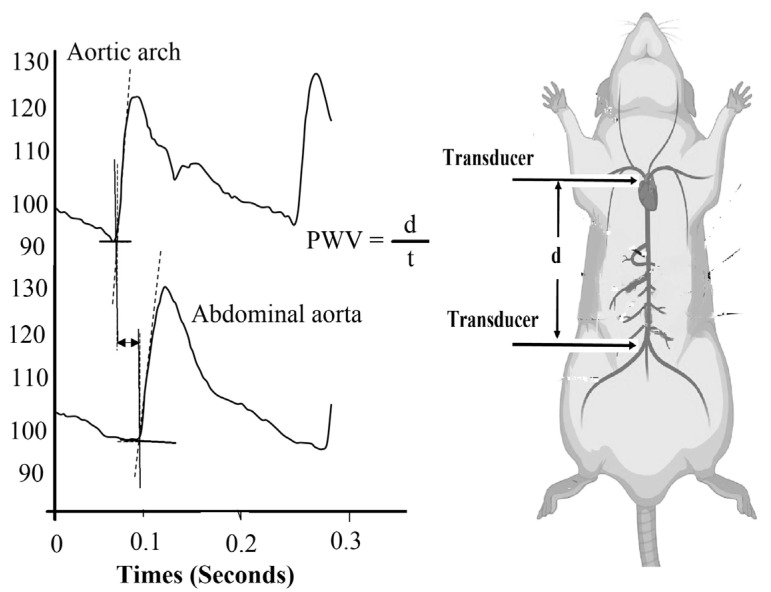
Schematic illustration of the measurement of pulse wave velocity (PWV) that is calculated by dividing the propagation distance (d) between tips of the two catheters by the propagation time (t) that is time difference at the minimal values of proximal (aortic arch) and distal (abdominal aorta) blood pressure.

**Figure 3 medicina-59-01880-f003:**
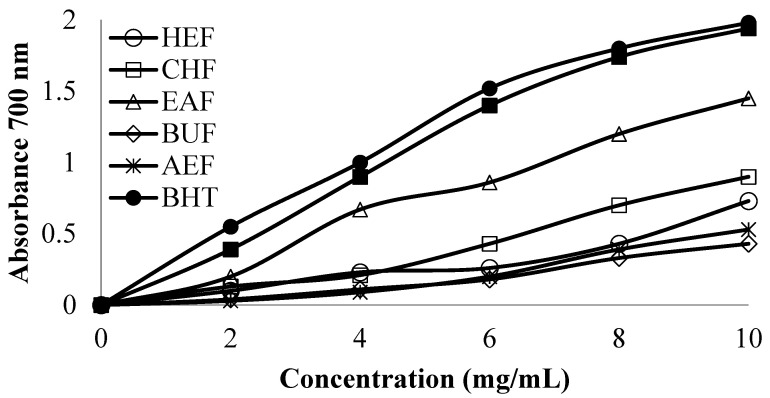
Reducing potential of polyphenol-rich *C. colocynthis* fruit fractions.

**Figure 4 medicina-59-01880-f004:**
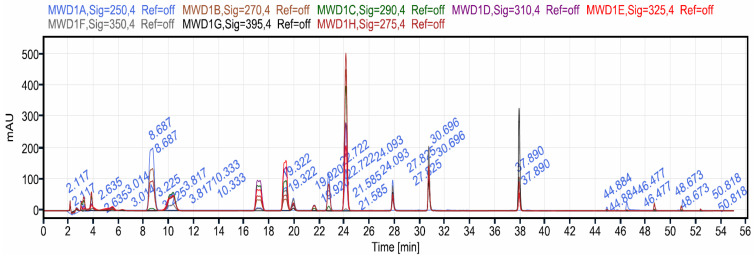
Typical HPLC chromatogram showing the separation of polyphenols using MWD.

**Figure 5 medicina-59-01880-f005:**
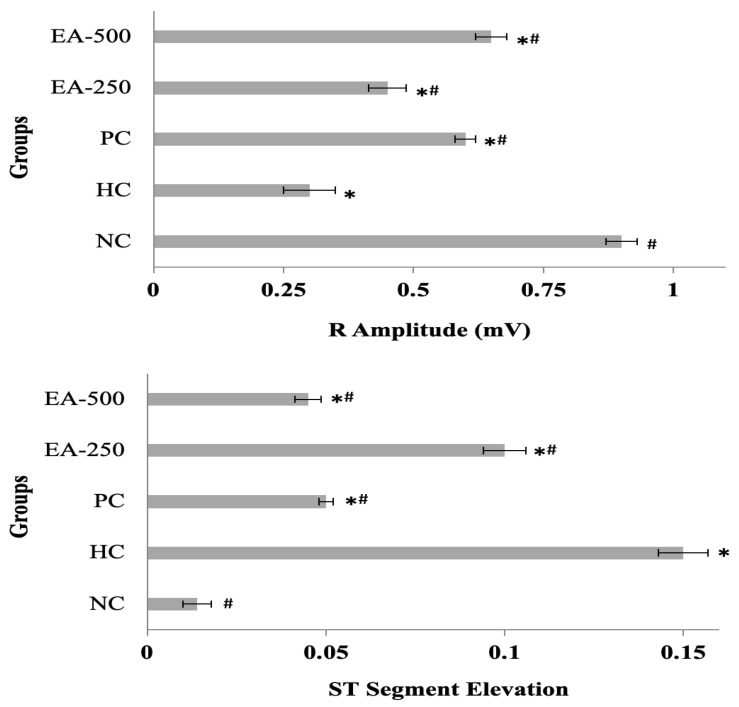
Changes in R-Amplitude and ST Segment in the ECG of different groups of spontaneously hypertensive rats. All data are presented as mean ± SD and were analyzed by one-way ANOVA followed by Bonferonni/Dunnett (all mean) post hoc test. NC, WHY normotensive control; HC, spontaneous hypertensive rats control; PC, positive control; EA-250 and EA-500, ethyl acetate fraction of *C. colocynthis* 250 and 500 mg/kg BW, respectively. * indicates the significant (*p* ≤ 0.05) difference of each treatment group from the NC group, # indicates the significant (*p* ≤ 0.05) difference from each treatment group from the HC group.

**Table 1 medicina-59-01880-t001:** Yield, spectrophotometric quantification of phenolic compounds, radical scavenging capacities and antioxidant activity of polyphenol-rich *C. colocynthis* fruit fractions.

Assays	HEF	CHF	EAF	BUF	AEF	BHT	BHA
Yields (g/100 g)	3.7 ± 0.1 ^b^	2.3 ± 0.1 ^a^	2.9 ± 0.2 ^a^	16.5 ± 0.8 ^c^	58.6 ± 1.5 ^d^	---	---
TPC (mg/g)	12.0 ± 1.0 ^a^	172.3 ± 4.0 ^d^	289.4 ± 5.1 ^e^	48.7 ± 2.2 ^c^	26.8 ± 1.1 ^b^		
TFC (mg/g)	0.6 ± 0.1 ^a^	3.5 ± 0.1 ^c^	7.6 ± 0.5 ^d^	2.5 ± 0.2 ^b^	2.1 ± 0.2 ^b^	---	---
FOL (mg/g)	1.9 ± 0.1 ^a^	10.2 ± 0.5 ^b^	35.7 ± 1.1 ^c^	9.5 ± 0.6 ^b^	9.3 ± 0.7 ^b^		
DPPH, SC_50_ (μg/mL)	115.9 ± 5.7 ^f^	17.3 ± 0.9 ^c^	6.2 ± 0.6 ^b^	22.4 ± 0.9 ^d^	53.7 ± 1.9 ^e^	5.2 ± 0.2 ^a^	5.0 ± 0.4 ^a^
ABTS, IC_50_ (μg/mL)	9782 ± 205 ^e^	298.7 ± 10 ^c^	79.5 ± 4.0 ^b^	590.3 ± 23 ^d^	671.1 ± 25	39.7 ± 2.1 ^a^	31.2 ± 2.0 ^a^
Inhibition of linoleic acid peroxidation (%)	56.1 ± 3.9 ^b^	79.9 ± 5.9 ^d^	83.1 ± 4.0 ^d^	67.2 ± 4.3 ^c^	47.9 ± 2.9 ^a^	88.4 ± 3.0 ^d^	89.8 ± 4.2 ^d^

Values are mean ± standard deviation of three independent experiments. Different letters in superscript in the same row represent significant differences (*p* ≤ 0.05) and various fractions. TPC, total phenolic content, mg/g of dry plant material, measured as gallic acid equivalent; TFC, total flavonoid content, mg/g of dry plant material, measured as catechin equivalent; FOL, flavonol content, mg/g of dry plant material, measured as rutin equivalent.

**Table 2 medicina-59-01880-t002:** Identification and quantification of polyphenols from ethyl acetate fraction of *C. colocynthis* fruit extract.

Compounds	T_R_ (Min)	Selected UV (nm)	(M + H) (*m*/*z*)	Concentration (mg/100 g of Plant Material)
Gallic acid	2.117	270	171	1.32 ± 0.03
*p*-Hydroxy-benzoic acid	2.63	270	139	1.81 ± 0.04
Epicatechin	3.225	270	291	2.05 ± 0.08
Caffeic acid	3.817	310	181	2.75 ± 0.07
*p*-coumaroylquinic acid	5.78	310	339	1.00 ± 0.03
Chlorogenic acid	8.687	325	355	18.93 ± 0.45
Vanillic acid	10.333	275	169	7.02 ± 0.34
Syringic acid	17.85	275	199	10.12 ± 0.44
Sinapic acid	19.322	325	225	14.51 ± 0.42
Ferulic acid	19.920	275	195	2.11 ± 0.10
Hesperidin	21.585	275	611	1.94 ± 0.09
Resveratrol	22.722	325	229	4.92 ± 0.14
Rutin	24.09	281	741	27.98 ± 1.45
Isoquercetin	27.825	250	465	3.14 ± 0.21
Kaempferol-3-glucoside	30.696	270	449	5.96 ± 0.20
Myricetin-3-O-glucuronide	37.890	270	495	9.21 ± 0.31
Myricetin-3-O-pentoside	44.884	270	451	1.09 ± 0.08
Quercetin-3-O-glucuronide	46.477	270	479	1.76 ± 0.07
Eriodictyol-7-O-rutinoside	48.673	290	597	1.23 ± 0.08
Apigenin glucoside	50.818	270	433	1.07 ± 0.06

Values are mean ± SD of triplicate determinations.

**Table 3 medicina-59-01880-t003:** Effect of ethyl acetate fraction of *C. colocynthis* on the body, kidney and liver weights and body mass, kidney and liver indexes of the different groups of the SHR model.

Groups	Body Weight (BW) (g)		Heart Weight (HW) (g)	HW/BW Ratio
	Initial	Final		
NC	135 ± 10 ^a^	238 ± 10 ^a^	0.80 ± 0.24 ^a^	0.336
HC	142 ± 14 ^a^	257 ± 12 ^a^	1.24 ± 0.17 ^a^	0.482
PC	145 ± 11 ^a^	243 ± 17 ^a^	0.87 ± 0.23 ^a^	0.358
EA-250	143 ± 12 ^a^	252 ± 11 ^a^	0.92 ± 0.20 ^a^	0.365
EA-500	144 ± 11 ^a^	250 ± 11 ^a^	0.89 ± 0.18 ^a^	0.356

The values are mean± standard deviation of six rats of the same group. Same letters in superscript in the same column show no significant (*p* > 0.05) differences among all the treatment and control groups. NC, WHY normotensive control; HC, spontaneous hypertensive rats control; PC, positive control; EA-250 and EA-500, ethyl acetate fraction of *C. colocynth* is 250 and 500 mg/kg BW, respectively.

**Table 4 medicina-59-01880-t004:** Cardiovascular parameters of different groups of the spontaneously hypertensive rat model.

Group	MAP (mmHg)	SBP (mmHg)	DBP (mmHg)	PP (mmHg)	PWV (m/s)	HR (bpm)
NC	108.9 ± 3.5 ^#^	128.1 ± 3.3 ^#^	92.0 ± 2.5 ^#^	33.9 ± 1.0 ^#^	5.33 ± 0.32 ^#^	332.4 ± 9.8
HC	154.2 ± 4.3 *	179.4 ± 4.1 *	138.6 ± 4.2 *	41.0 ± 1.5 *	7.12 ± 0.64 *	347.3 ± 9.5
PC	121.2 ± 3.7 *^#^	140.3 ± 4.3 *^#^	106.7 ± 3.0 *^#^	36.1 ± 0.9 *^#^	5.89 ± 0.38 ^#^	340.6 ± 9.0
EA-250	129.3 ± 3.8 *^#^	151.5 ± 6.4 *^#^	119.5 ± 4.7 *^#^	38.2 ± 0.8 *^#^	6.21 ± 0.40	343.0 ± 9.7
EA-500	119.8 ± 5.7 *^#^	139.8 ± 3.4 *^#^	107.2 ± 2.7 *^#^	35.8 ± 1.3 ^#^	5.80 ± 0.42 ^#^	336.0 ± 9.9

All the data are presented as mean ± SD and were analyzed by one-way ANOVA followed by Bonferonni/Dunnett (all mean) post hoc test. NC, WHY normotensive control; HC, spontaneous hypertensive rats control; PC, positive control; EA-250 and EA-500, ethyl acetate fraction of *C. colocynthis* 250 and 500 mg/kg BW, respectively. * indicates a significant (*p* ≤ 0.05) difference from the NC group; ^#^ indicates a significant (*p* ≤ 0.05) difference from the HC group.

**Table 5 medicina-59-01880-t005:** Oxidative stress parameters of different groups of the spontaneously hypertensive rat model.

Group	MDA (nmol/L)	GSH (mg/L)	SOD (NU/mL)	NOx (nmol/mL)	TAC (mmol/L)
NC	2.81 ± 0.14 ^#^	163.2 ± 6.2 ^#^	139.6 ± 6.1 ^#^	31.95 ± 1.20 ^#^	1.78 ± 0.09 ^#^
HC	5.79 ± 0.23 *	124.3 ± 4.1 *	117.0 ± 4.2 *	19.13 ± 0.49 *	1.31 ± 0.06 *
PC	2.90 ± 0.18 ^#^	159.7 ± 4.2 ^#^	133.9 ± 4.0 ^#^	30.05 ± 0.99 ^#^	1.73 ± 0.10 ^#^
EA-250	4.00 ± 0.17 *^#^	142.9 ± 4.3 *^#^	126.8 ± 3.7 *^#^	28.89 ± 1.01 *^#^	1.72 ± 0.10 ^#^
EA-500	2.91 ± 0.11 ^#^	161.0 ± 6.0 ^#^	135.7 ± 3.5 ^#^	30.09 ± 1.03 ^#^	1.79 ± 0.07 ^#^

All data are presented as mean ± SD and were analyzed by one-way ANOVA followed by Bonferonni/Dunnett (all mean) post hoc test. NC, WHY normotensive control; HC, spontaneous hypertensive rats control; PC, positive control; EA-250 and EA-500, ethyl acetate fraction of *C. colocynthis* 250 and 500 mg/kg BW, respectively. * indicates a significant (*p* ≤ 0.05) difference from the NC group; ^#^ indicates a significant (*p* ≤ 0.05) difference from the HC group.

## Data Availability

Data available on request from the authors.

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
