# Peer review of "Bioactivity-Guided Isolation and Antihypertensive Activity of Citrullus colocynthis Polyphenols in Rats with Genetic Model of Hypertension"

_medicina, 2023, doi:10.3390/medicina59101880_

Round 1

Reviewer 1 Report

Dear Authors,

Here are my comments and suggestions:

1.       Line 11: Please put a brief introduction on Citrullus colocynthis poly-2phenols

2.       Line 12: I would suggest to delete this “in rats with genetic model of hypertension called as”.

3.       Line 13: I would suggest to delete this “Soxhlet extraction technique is used to prepare ethanol extract of CC.” Detailed extraction method should be explained in the methodology section.

4.       Line 16: spelling of “toal”

5.       Line 305: spelling of “C. colucynthis”

6.       Please make sure to italicized C. colocynthis in the whole manuscript.

7.       line 85: “not rarely reported”. I would suggest to change the word not rarely into its synonym such as commonly/frequently.

8.       Line 86-97 : Please re-do this paragraph. Just state the aim. Do not mention about the methodology. My suggestion:  “ Therefore, in this present study, we aimed to determine the effects of polyphenol-rich fractions of C. colocynthis fruits that were prepared using various solvents for antioxidant and antihypertensive activities” Line 87-97 can be deleted, as all these will be included in the methodology part.

9.       Line 234 and 501: change “Dozes” to “Doses”

10.   Line 603: EA-250 and EZ-500 groups. Change EZ to EA

11.   Line 647: add of CC. “EAF of CC”

Thank you.

I have detected some spelling and grammar mistakes. Please correct them accordingly. Thank you.

Author Response

Response to Reviewers

The manuscript (medicina-2627924) has been carefully revised in all aspects. The comments from all five reviewers have been carefully addressed. A number of additional improvements have been made, regarding grammar, clarity and language. All the new changes/amendments have been marked by blue color.

Reviewer 1

Comment 1: Line 11: Please put a brief introduction on Citrullus colocynthis polyphenols.

Response: A brief introduction of Citrullus colocynthis is added at the start of the abstract.

Comment 2: Line 12: I would suggest to delete this “in rats with genetic model of hypertension called as”.

Response: The sentence has been revised as suggested.

Comment 3: Line 13: I would suggest to delete this “Soxhlet extraction technique is used to prepare ethanol extract of CC.” Detailed extraction method should be explained in the methodology section.

Response: The line is deleted from the abstract. The extraction detail is already given in the methodology section.

Comment 4: Line 16: spelling of “toal”

Response: The correction has been made. 

Comment 5: Line 305: spelling of “C. colucynthis”

Response: Typographic mistake has been rectified.

Comment 6: Please make sure to italicized C. colocynthis in the whole manuscript.

Response: The manuscript has been carefully revised and ensured that all the scientific names given are now in italic font.

Comment 7: line 85: “not rarely reported”. I would suggest to change the word not rarely into its synonym such as commonly/frequently.

Response: Sentence has been revised.

Comment 8: Line 86-97 : Please re-do this paragraph. Just state the aim. Do not mention about the methodology. My suggestion:  “ Therefore, in this present study, we aimed to determine the effects of polyphenol-rich fractions of C. colocynthis fruits that were prepared using various solvents for antioxidant and antihypertensive activities” Line 87-97 can be deleted, as all these will be included in the methodology part.

Response: Thanks for the suggestion. The sentence has been revised accordingly and additional lines have been deleted.

Comment 9: Line 234 and 501: change “Dozes” to “Doses”

Response: Typographic mistakes have been corrected.

Comment 10: Line 603: EA-250 and EZ-500 groups. Change EZ to EA

Response: Correction has been made.

Comment 11: Line 647: add of CC. “EAF of CC”

Response: Thanks. The correction has been done.

Reviewer 2 Report

The research work favors in an important way a study that contributes important elements with antihypertensive and antioxidant activities, it is important to consider and note in the study that although fractions of compounds were separated in most of the times the effect is due to the effect that can generate the drug in toto, although there are majority compounds in the fractions. Therefore it is important to discuss in the article in this regard.

 It should also discuss the need for natural pharmacological interactions between the components in a plant to achieve the desired effect, since there are few plants that have leading molecules capable of generating therapeutic effects on their own, used as an option.

Also, you must first define your abbreviations to be able to use them in the paper, there are several abbreviations not defined in the document.

Author Response

Response to Reviewers

The manuscript (medicina-2627924) has been carefully revised in all aspects. The comments from all five reviewers have been carefully addressed. A number of additional improvements have been made, regarding grammar, clarity and language. All the new changes/amendments have been marked by blue color.

Reviewer 2

General Comment: The research work favors in an important way a study that contributes important elements with antihypertensive and antioxidant activities, it is important to consider and note in the study that although fractions of compounds were separated in most of the times the effect is due to the effect that can generate the drug in toto, although there are majority compounds in the fractions. Therefore it is important to discuss in the article in this regard.  It should also discuss the need for natural pharmacological interactions between the components in a plant to achieve the desired effect, since there are few plants that have leading molecules capable of generating therapeutic effects on their own, used as an option. Also, you must first define your abbreviations to be able to use them in the paper, there are several abbreviations not defined in the document.

Response: Thanks for the suggestions. The manuscript is carefully revised and the full names followed by abbreviations have been given now. The discussion is improved and the interactions of compounds from the fractions have been discussed now.   

Reviewer 3 Report

Dear Author/s,

I review the paper carefully. It explores the antihypertensive and antioxidant potentials of Citrullus colocynthis (CC) polyphenols using a genetic model of hypertension in rats called spontaneous hypertensive rats (SHR). The polyphenol-rich fractions of CC fruits were prepared using various solvents to evaluate their antioxidant and antihypertensive activities. The study characterizes the phytochemical components of the polyphenol-rich fraction using HPLC-MWD-ESI-MS analysis, identifying twenty polyphenol compounds, including phenolic acids and flavonoids, mainly myricetin and quercetin derivatives. In vivo experiments on SHR showed that the ethyl acetate fraction (EAF) of CC exhibited significant antihypertensive activity by reducing mean arterial pressure, systolic blood pressure, diastolic blood pressure, and pulse pressure. EAF also demonstrated strong antioxidant activity, as evidenced by its ability to scavenge DPPH and ABTS radicals and normalize oxidative stress markers in the animals. Additionally, the treatment with EAF improved the electrocardiogram pattern and pulse wave velocity in the animals.

I just added a suggestion at the end of the "Introduction" section. Please, check the attached file and revise it. 

Good Luck

Author Response

Response to Reviewers

The manuscript (medicina-2627924) has been carefully revised in all aspects. The comments from all five reviewers have been carefully addressed. A number of additional improvements have been made, regarding grammar, clarity and language. All the new changes/amendments have been marked by blue color.

Reviewer 3

Comment: I review the paper carefully. It explores the antihypertensive and antioxidant potentials of Citrullus colocynthis (CC) polyphenols using a genetic model of hypertension in rats called spontaneous hypertensive rats (SHR). The polyphenol-rich fractions of CC fruits were prepared using various solvents to evaluate their antioxidant and antihypertensive activities. The study characterizes the phytochemical components of the polyphenol-rich fraction using HPLC-MWD-ESI-MS analysis, identifying twenty polyphenol compounds, including phenolic acids and flavonoids, mainly myricetin and quercetin derivatives. In vivo experiments on SHR showed that the ethyl acetate fraction (EAF) of CC exhibited significant antihypertensive activity by reducing mean arterial pressure, systolic blood pressure, diastolic blood pressure, and pulse pressure. EAF also demonstrated strong antioxidant activity, as evidenced by its ability to scavenge DPPH and ABTS radicals and normalize oxidative stress markers in the animals. Additionally, the treatment with EAF improved the electrocardiogram pattern and pulse wave velocity in the animals.

I just added a suggestion at the end of the "Introduction" section. Please, check the attached file and revise it. (The last sentence of the introduction, usually pointed to the main general aim of the whole paper not detailed method. I think this information should be explained in method section not at the end of introduction.)

Response: Thanks for the suggestion. The sentence regarding the aim of the study has been revised accordingly and additional lines related to the methodology have been removed from the last paragraph of the introduction.

Reviewer 4 Report

the article titled "Bioactivity-guided isolation and antihypertensive activity of Citrullus colocynthis poly- 2 phenols in rats with genetic model of hypertension" has a good subject. the author of this article tries to evaluate the antihypertensive activity of Citrullus colocynthis. the article needs some corrections. 

1- The introduction is poorly written and must include c.colocynhis ingredients. which ingredient has antihypertensive must discussed. in addition, possible mechanisms of action should be explained.

2- Please add the chemical and reagent section to the method section.

3- Author contribution not mentioned

4- The statement needs to be removed.

5- methods section; please add relevant references to each method. please add SPSS program details to the statistical analysis section.

6- it is obvious the statistical analysis between NC and HC needs to be repeated (fig 5).

7- Please repeat all SPSS analyses and provide raw data as supplementary files. 

Author Response

Response to Reviewers

The manuscript (medicina-2627924) has been carefully revised in all aspects. The comments from all five reviewers have been carefully addressed. A number of additional improvements have been made, regarding grammar, clarity and language. All the new changes/amendments have been marked by blue color.

Reviewer 4

The article titled "Bioactivity-guided isolation and antihypertensive activity of Citrullus colocynthis polyphenols in rats with genetic model of hypertension" has a good subject. The author of this article tries to evaluate the antihypertensive activity of Citrullus colocynthis. The article needs some corrections. 

Comment 1: The introduction is poorly written and must include C.colocynhis ingredients. which ingredient has antihypertensive must discussed. In addition, possible mechanisms of action should be explained.

Response: The introduction section is revised and more data regarding the C. colocynthis bioactive compounds related to cardioprotective and antioxidant study are added. Furthermore, the discussion section is improved and possible mechanism of action is also discussed.

Comment 2: Please add the chemical and reagent section to the method section.

Response: Chemicals and reagents section is added as subsection 2.1.

Comment 3: Author contribution not mentioned

Response: Author contributions are now mentioned in the revised version of the manuscript.

Response: Authors contribution is added now in the relevant section.

Comment 4: The statement needs to be removed.

Response: Removed.

Comment 5: Methods section; please add relevant references to each method. Please add SPSS program details to the statistical analysis section.

Response: The relevant references of each method have been provided. The statistical program details have also been added in the statistical analysis section.

Comment 6: It is obvious the statistical analysis between NC and HC needs to be repeated (fig 5).

Response: Thanks to point out. The statistical analysis is rechecked. Actually the “*” indicates significant (p ≤ 0.05) difference of each treatment group from the normal control (NC) group, and “#” indicates significant (p ≤ 0.05) difference of each treatment group from the hypertensive control (HC) group. The sentence in figure footnote is revised to make it more clear.

Comment 7: Please repeat all SPSS analyses and provide raw data as supplementary files. 

Response: All the in vitro analyses were repeated three times and the data is reported as mean value ± standard deviation. For in vivo analysis, 6 rats were taken in each group and results are reported as mean ± standard deviation. To compare the difference between values, one way Analysis of Variance (ANOVA) followed by Bonferroni/Dunnett (all mean) post hoc test were applied using statistical software (Statistica; Stat Sift Inc, Tulsa, OK, USA). The probability value (p) ≤ 0.05 were considered significantly different. The supplementary files are now uploaded.

Reviewer 5 Report

Dear authors,

I have read with interest your paper, which I found constructed as a fairly conducted pre-clinical study. Still, I have some issues to address:

1. The citing of references in the text does not comply with the journal`s section ”Instruction for authors”. Please use specific parentheses and change the style of reference reporting. 

2. On a quick search, I found these errors only in the abstract:

"toal" in line 16 should be "total."

"shower" in line 21 should be "showed."

"doze" in line 27 should be "dose."

"EA-500" in line 30 should be "EAF-500" to maintain consistency with the previous abbreviation.

A description of EAF is made, but EA-500 is not explained in the Abstract. The EA-250 and EA-500 must be explained in the text and abbreviated as EAF-250/500 for consistency.

There are many other small typing errors, you should get the text checked again.

3. The figures (1 and 2) look like they were scanned from old books, with a low quality of presentation. Please try to update the figures to be more clear and neat. 

4. C. colocynthis should be written in Italic font throughout the text. 

5. I would advise separating the Results and the Discussion chapters, as the Discussion section could become richer if you would properly and extensively discuss your results.

Good luck!

There are some typing errors in the text. 

Author Response

Response to Reviewers

The manuscript (medicina-2627924) has been carefully revised in all aspects. The comments from all five reviewers have been carefully addressed. A number of additional improvements have been made, regarding grammar, clarity and language. All the new changes/amendments have been marked by blue color.

Reviewer 5

I have read with interest your paper, which I found constructed as a fairly conducted pre-clinical study. Still, I have some issues to address:

Comment 1: The citing of references in the text does not comply with the journal`s section “Instruction for authors”. Please use specific parentheses and change the style of reference reporting. 

Response: The citations of references in the text have been corrected as per journals instruction now.

Comment 2: On a quick search, I found these errors only in the abstract: "toal" in line 16 should be "total.", "shower" in line 21 should be "showed.", "doze" in line 27 should be "dose.", EA-500" in line 30 should be "EAF-500" to maintain consistency with the previous abbreviation., A description of EAF is made, but EA-500 is not explained in the Abstract. The EA-250 and EA-500 must be explained in the text and abbreviated as EAF-250/500 for consistency. There are many other small typing errors, you should get the text checked again.

Response: The manuscript has been revised carefully and all the typographical, grammar and spelling mistakes have been corrected. Some sentences have been revised for clarity. All the abbreviations have been defined, when appeared first time in the text. All the new changes/amendments have been marked by blue color.

Comment 3: The figures (1 and 2) look like they were scanned from old books, with a low quality of presentation. Please try to update the figures to be more clear and neat. 

Response: No these were not scanned copies. These figures were made by author using paint application. The sizes of the pictures are increased to make them clearer.

Comment 4: C. colocynthis should be written in Italic font throughout the text. 

Response: The manuscript has been carefully revised and ensured that all the scientific names given are now in italic font.

Comment 5: I would advise separating the Results and the Discussion chapters, as the Discussion section could become richer if you would properly and extensively discuss your results.

Response: Discussion section is improved and separated and more detail is added along with more literature.

Round 2

Reviewer 4 Report

Dear Editor

The author did respond to all questions in the text. It is ready and accepted.